# The Influence of the COVID-19 Pandemic in NK Cell Subpopulations from CML Patients Enrolled in the Argentina Stop Trial

**DOI:** 10.3390/cells14090628

**Published:** 2025-04-23

**Authors:** María Belén Sanchez, Bianca Vasconcelos Cordoba, Carolina Pavlovsky, Beatriz Moiraghi, Ana Ines Varela, Isabel Giere, Mariana Juni, Nicolas Flaibani, José Mordoh, Julio Cesar Sanchez Avalos, Estrella Mariel Levy, Michele Bianchini

**Affiliations:** 1Centro de Investigaciones Oncológicas–Fundación Cáncer FUCA, Buenos Aires 1426, Argentina; biancavcordoba@gmail.com (B.V.C.); josemordoh39@gmail.com (J.M.); estrellamlevy@yahoo.com.ar (E.M.L.); 2Fundación para Combatir la Leucemia (FUNDALEU), Buenos Aires 1114, Argentina; cpavlovsky@fundaleu.org.ar (C.P.); igiere@fundaleu.org.ar (I.G.); mjuni@fundaleu.org.ar (M.J.); 3Hospital José María Ramos Mejía, Buenos Aires 1221, Argentina; beatrizmoiraghi@hotmail.com (B.M.); anainesvarelap@gmail.com (A.I.V.); 4Laboratorio de Evolución, Facultad de Ciencias Exactas y Naturales (FCEN-UBA), Instituto de Genética, Ecología y Evolución de Buenos Aires (IEGEBA), Buenos Aires 1428, Argentina; n.flaiba@gmail.com; 5Instituto Alexander Fleming, Buenos Aires 1426, Argentina; jcsanchezavalos@yahoo.com.ar

**Keywords:** NK cells, COVID, chronic myeloid leukemia, treatment-free remission

## Abstract

Treatment-free remission (TFR) is a key therapeutic goal for chronic myeloid leukemia (CML) patients in deep molecular response (DMR). While predicting patient outcome remains challenging, different NK cell populations seem crucial. We conducted an immunological sub-study from the Argentina Stop Trial (AST), including 46 patients in 2019 (AST I) and 35 new patients between 2022 and 2023 (AST II). To characterize NK cell subsets in patients attempting TFR, peripheral blood mononuclear cell samples were collected before stopping treatment and phenotype and functional characteristics were assessed by flow cytometry. Non-relapsing patients from AST I exhibited NK cell subpopulations with cytomegalovirus-related memory features, high expression of cytotoxicity markers, and robust functionality. Remarkably, though clinical variables were very similar between cohorts, significant immune differences were observed. NK cell percentage and CD16 and CD57 receptor expression levels were significantly reduced in AST II (*p* = 0.0051; *p* = 0.0222; *p* = 0.0033, respectively), whereas NKp46, NKp44 and PD-1 expression levels were significantly increased (*p* = 0.0081; *p* < 0.0001; *p* < 0.0001, respectively). NK cells from AST II patients demonstrated higher overall functionality and more memory-like subpopulations, characterized mainly by the expression of CD57, NKG2C, NKp30 and NKp46 receptors among CD56^dim^ NK cells, also with enhanced functional performance. However, in AST II, we were unable to report an association with clinical outcome. Given the enrollment time of both cohorts and that they appear to be clinically homogeneous, we consider that COVID could be impacting the immune landscape; accordingly, serum samples from AST II, but not AST I, confirmed the presence of anti-SARS-CoV-2 IgG. The influence of the COVID pandemic and the different vaccine platforms on NK cells cannot be underestimated when evaluating the role of the immune system in cancer.

## 1. Background

Chronic myeloid leukemia (CML) is a myeloproliferative neoplasm caused by the chimeric oncogene BCR-ABL1 [1]. Tyrosine kinase inhibitors (TKIs) represent the therapeutic standard [2,3,4,5] and those patients who achieve and sustain deep molecular response (DMR) are candidates for discontinuing therapy, establishing treatment-free remission (TFR) as a therapeutic goal. Approximately half of patients sustain TFR [6,7,8,9], and it is known that, even then, leukemic stem cells could persist [10].

It is believed that this reservoir could be controlled by the immune system, mainly by Natural Killer (NK) cells [11,12,13]. In recent years, there has been growing interest in subpopulations of NK cells with adaptive features, mainly induced by a previous infection with human cytomegalovirus (HCMV) [14,15,16]. In the context of hematological malignancies, several reports found a lower risk of relapse after hematopoietic cell transplantation and better disease-free survival when HCMV reactivation occurred, in both CML and acute myeloid leukemia (AML) [17,18]. Subsequently, this was associated with the presence of a subpopulation of CD56^dim^CD57^+^NKG2C^+^ NK cells that expanded in response to the early reactivation of HCMV post-transplant, which was not detected in HCMV^+^ recipients without virus reactivation or in HCMV^−^ recipients. In addition, this subpopulation exhibited increased production of TNFα and IFNγ; [19,20].

Working with a first cohort (AST I: 2019) of HCMV^+^ patients, we found that those who can sustain TFR present a greater proportion of adaptive-like NK subpopulations, characterized by the expression of the markers NKG2C and CD57, but also with high expression of natural cytotoxicity receptors NKp30 and NKp46, something that is not common in adaptive-like subpopulations. Furthermore, it was observed that, although the degranulation capacity was equivalent in both response groups, the greater proportion of this subset within the CD56^dim^ cells of patients who did not relapse resulted in higher levels of degranulation, suggesting a protective role of these cell subsets in this clinical context [21]. To validate these findings, we recruited a second cohort of patients (AST II: 2022–2023) from whom we obtained peripheral blood samples at baseline to characterize the phenotype and functionality of these subsets and evaluate whether they are associated with maintenance of response.

## 2. Materials and Methods

We conducted an immunological sub-study from the Argentina Stop Trial (AST), including 46 patients in 2019 (AST I) and 35 new patients between 2022 and 2023 (AST II), from which peripheral blood samples were collected before stopping treatment. All patients gave written informed consent. First, samples were centrifuged to obtain plasma, which was stored at −80 °C until its use to determine HCMV and SARS-CoV-2 seropositivity. Peripheral blood mononuclear cells (PBMCs) were isolated using a Ficoll-Paque Plus density gradient (GE Healthcare, Chicago, IL, USA) following the manufacturer’s instructions, washed with saline solution, and immediately used for flow cytometry or cryopreserved in freezing medium (90% FBS (Gibco, Grand Island, NY, USA), 10% DMSO). Phenotype was assessed by flow cytometry (BD FACS Canto™II; Eysins, Switzerland), and the remaining cells were cultured with the K562 cell line at a 1:5 NK:K562 cell ratio for 6 h at 37 °C to measure degranulation and IFNγ production (Figure 1).

Anti-HCMV IgG were quantitatively determined from plasma samples through the ELFA technique (enzyme-linked fluorescence assay), using the VIDAS CMV IgG kit (Biomerieux, Lyon, France). On the other hand, anti-SARS-CoV-2 IgG were determined semiquantitatively from plasma samples using the COVIDAR IgG ELISA kit (CONICET, Fundación Instituto Leloir, Universidad de San Martín and Laboratorio Lemos S.R.L.), following the manufacturer’s instructions.

Flow cytometry data were analyzed using FlowJo software version number 10.6.2 and GraphPad Prism version number 9 was used for statistical analysis. The non-parametric Mann–Whitney test was used for comparing differences between groups and for AST I vs. AST II comparisons, and *p*-values were corrected using the False Discovery Rate method (FDR–Benjamini–Hochberg) (Appendix A). Molecular recurrence-free survival was estimated by the Kaplan–Meier method and compared within groups by the log-rank test. Differences were considered statistically significant when *p* < 0.05.

## 3. Results

In the AST I cohort, there were 18 relapsed patients, mostly within the first 6 months (13/18), with 2 relapses occurring between Month 6 and Month 12, 1 relapse occurring at Month 21, and the remaining 2 relapses occurring almost 3 and a half years after discontinuing treatment. Meanwhile, for the AST II cohort, there were 13 relapses, all within the first 4 months post-discontinuation. The original follow-up period for each cohort was two years, but it is worth noting that in the case of the first cohort, we have data on later relapses because most patients continued on molecular monitoring.

Following results from AST I, in order to validate those immunological findings, we decided to study a second cohort of patients (AST II: 2022–2023). Surprisingly, even though the inclusion criteria were broadly the same, no significant differences were observed among the different adaptive-like NK cell subsets in terms of proportion (Figure 2) and functionality (Figure 3) between relapsed vs. non-relapsed patients, except for IFNγ levels and relapse-free survival (RFS), which even goes against what was previously reported in our study.

To understand the origin of these differences, we first decided to compare several clinical variables between the two cohorts of patients such as months under treatment, age, months in DMR before discontinuation, age at diagnosis, proportion of Sokal risk and type of TKI. As shown in Figure 4, none of these variables reached significant differences.

Instead, cohorts turned out to be very different from each other at an immunological level, showing a significant decrease in the percentage of total NK cells and in the proportion of CD57 and CD16 receptors for AST II patients (Figure 5A–C), while a significant increase was observed in the expression of NKp46, NKp44 and PD-1 (Figure 5D–F). Overall, these findings suggest a less mature but more activated phenotype for NK cells from AST II patients. Furthermore, all adaptive-like NK cell subpopulations showed significantly increased proportions in AST II (Figure 6), and the functional performance was also systematically increased in the second cohort of patients when evaluating these parameters both at a global level (Figure 7) and among the CD56^dim^NKG2C^+^NKp46^+^ subpopulation (Figure 8). Lastly, taking into account the fact that memory-associated subpopulations expand as a result of HCMV infection, we wondered if there were differences in the antibody titer between groups, but we did not find significant differences either (Appendix A), suggesting that the functional difference would not be entirely related to this parameter.

Finally, even though RFS at one year was similar for both cohorts (67% for AST I vs. 65% for AST II, Figure 9A,C), it is striking how 100% of AST II relapses were concentrated in the first 4 months after discontinuation, while for AST I, only 40% of relapses occurred within the same period of time (Figure 9B,D).

How could we explain the difference at the immunological level between the two groups, even though no significant changes were evident at the clinical level? To answer this question, it is important to consider that all samples from AST I patients were obtained prior to the local emergence of the SARS-CoV-2 pandemic (February 2019 to February 2020). Meanwhile, the second cohort was recruited between July 2022 and April 2023, a time when, whether due to natural infection or current vaccination schedule, most individuals had already suffered some type of exposure to the virus. To verify this, we decided to use part of the cryopreserved plasma samples to test for SARS-CoV-2 seropositivity by ELISA. Samples from the entire AST II cohort were used along with 12 randomly selected control cases from the AST I cohort. This showed, as expected, that none of the patients in the first cohort presented antibodies against SARS-CoV-2 while 94% of the AST II patients were positive, the vast majority presenting a high titer, with only two seronegative patients (Figure 9E).

## 4. Discussion

The mechanisms behind successful TFR are still not entirely clear, but multiple sources of evidence suggest that NK cells may play a key role in controlling residual disease [11,12,13]. The objective of the AST II study was to evaluate the feasibility of discontinuing therapy in patients who met the requirements and, in addition, to search for possible biomarkers that could predict the clinical outcome of the patients, or in any case, identify the biological differences between those who maintain the response and those who suffer a relapse of the disease.

From the AST I cohort, we were able to characterize a subset of adaptive-like NK cells with overall increased degranulation capacity, which appears to confer a certain level of protection for patients who do not relapse [21]. Since further studies in larger cohorts were needed, we recruited more patients (AST II) to test this hypothesis. Although the inclusion criteria were the same for both groups, and these turned out to be clinically very homogeneous, the relapse rate of AST II was much faster than that of the first cohort and the protective role of adaptive subpopulations was not so clear for AST II patients, where even those who relapsed presented high proportions of these subsets. Here, we could speculate that although these subpopulations are present at high levels, and appear to be functionally active, something prevents them from recognizing and killing residual tumor cells. While not seemingly exhausted, NK cell degranulation and IFNγ production do not appear to fully capture the complexity of their interaction with leukemic stem cells (LSCs). Unfortunately, residual LSCs are very difficult to detect and characterize in terms of surface ligand expression, so we recognize that further research would be needed to explore how they might influence NK cell responses in CML.

When looking into immunological differences between both cohorts, not only did NK cells from AST II patients appear to be less mature but more activated, but also, every adaptive-like subset was significantly increased, and both global and adaptive functionality were exacerbated for the second cohort, when HCMV titers did not show significant differences. In this scenario, our hypothesis is that the COVID-19 pandemic could be a possible explanation since, beyond their antitumor activity, NK cells are also essential for the control of viral infections. In SARS-CoV-2 infection, it has been reported that the number of NK cells decreases, a phenomenon that directly correlates with the severity of the disease [22]. This is consistent with our observations of a lower percentage of NK cells in AST II compared to AST I. Similarly, PD-1 is another of the markers that was reported to be increased in the NK cells of COVID^+^ patients [23]; accordingly, we also observed a significant increase in AST II patients. On the other hand, as mentioned above, HCMV infection gives rise to a particular subtype of mature NK cells (CD57^+^) further characterized by the expression of the NKG2C receptor. This also occurs after infection with other viruses, including SARS-CoV-2. Indeed, various authors report the presence of CD57^+^NKG2C^+^ NK cells in convalescent donors, indicating the presence and expansion of adaptive-like NK cells [24]. This phenomenon could explain why we observed a significant increase in AST II for adaptive-like NK cells, despite no significant differences being observed for HCMV titer between the two cohorts.

Not only natural infection generates changes in the immune system, but also vaccination. In healthy donors without a previous positive COVID diagnosis, vaccinated with two doses of Pfizer–BioNTech (BNT162b2) plus a booster dose, considerable differences were found in the expression of certain markers one month after the booster dose, including a high level of NKG2C and CD57 within the CD56^dim^CD16^+^ subset. Furthermore, the percentages of CD56^dim^CD16^+^NKG2C^+^ and CD56^dim^CD16^+^CD57^+^ cells were significantly correlated with the IgG titer against SARS-CoV-2 [25].

Furthermore, a recent report has shown that infection with SARS-CoV-2 leads to increased HLA-E expression in lung cells, which in turn results in enhanced degranulation, IFNγ production and target cell cytotoxicity in NKG2C+ adaptive-like NK cells. This suggests a memory generation mechanism in common with that induced by HCMV infection, but at the same time independent [26].

These reports support the hypothesis that COVID-19 infection may have altered the expression of multiple markers associated with adaptive-like NK cells. This effect could explain the discrepancies observed between the two cohorts in terms of immunological profile and clinical outcome. Therefore, we believe that all the observations from the AST II cohort are more related to the influence of SARS-CoV-2 rather than the underlying pathology or the clinical context of therapy discontinuation. It would be valuable to explore differences in outcomes between patients who developed immunity through both infection and vaccination compared to those who were vaccinated alone, and also between different vaccine platforms. Unfortunately, we do not have that information for our patients currently, but we recognize it as a significant area for future research once the necessary data become accessible. On the other hand, we acknowledge that the small sample size from each cohort (46 patients in AST I and 35 patients in AST II) could limit the statistical power of the study and the generation of robust conclusions. However, these observations still provide valuable insights into the potential impact of SARS-CoV-2 on NK cell function in CML patients; this is why we recommend that further research in larger cohorts could take them into account, and test them, in order to validate and ideally better understand our findings.

## 5. Conclusions

The primary objective of expanding the patient cohort of this discontinuation trial was to establish the role of adaptive-like NK cells in maintaining TFR. However, in the second cohort, limited conclusions could be drawn to answer the original question about what differentiates a patient who relapses from one who sustains TFR. Instead, observations suggest a direct impact of SARS-CoV-2 on NK cell immunophenotype and function. It would be convenient to deepen the knowledge related to both the role of adaptive-like NK cells in patients who attempt therapy discontinuation, as well as the influence of COVID on the generation and maintenance of these subpopulations.

## Figures and Tables

**Figure 1 cells-14-00628-f001:**
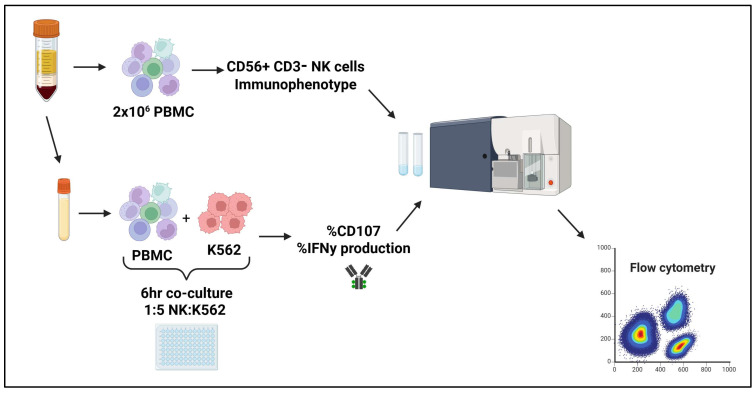
Experimental design.

**Figure 2 cells-14-00628-f002:**
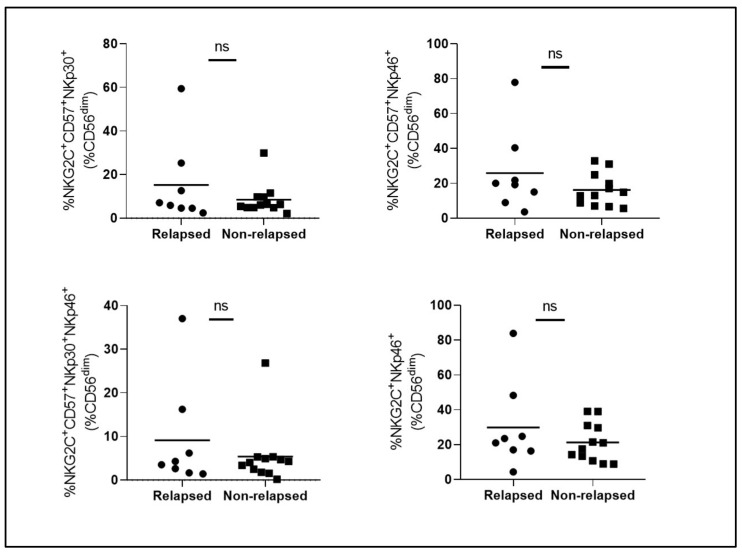
Comparison of the proportion of different adaptive-like NK cell subsets between relapsed vs. non-relapsed patients from the AST II cohort. Circles represent relapsed patients. Squares represent non-relapsed patients. ns: non-significant.

**Figure 3 cells-14-00628-f003:**
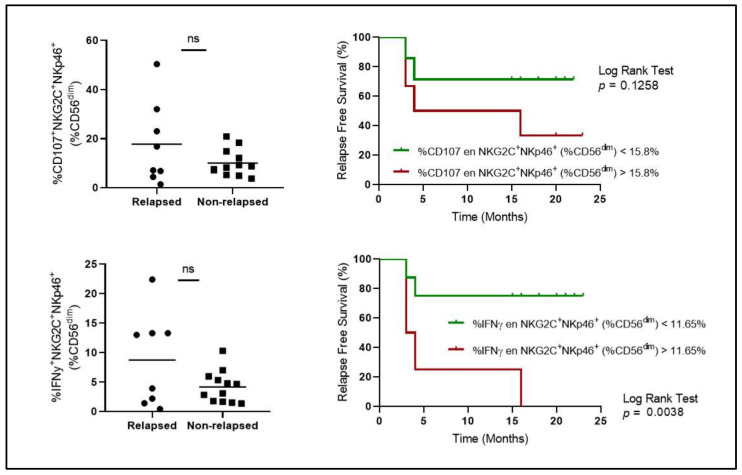
Degranulation (%CD107a) and IFNγ production of relapsed vs. non-relapsed patients from AST II cohort. ns: non-significant.

**Figure 4 cells-14-00628-f004:**
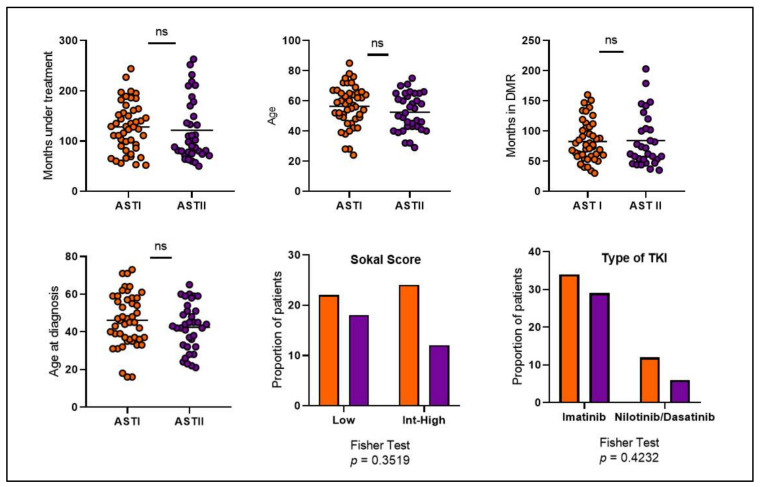
Comparison of various clinical variables between both patient cohorts. Orange circles and bars represent AST I cohort. Violet circles and bars represent AST II cohort. Int (Intermediate). ns: non-significant.

**Figure 5 cells-14-00628-f005:**
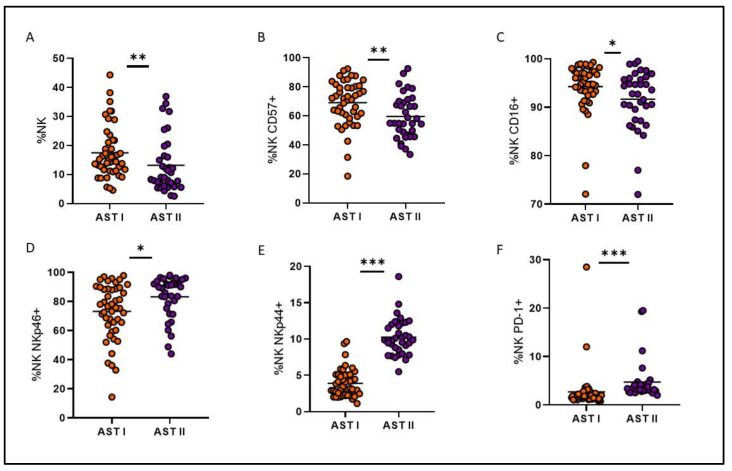
Comparison of the proportion of total NK cells (**A**) and several NK cell receptors between both patient cohorts (**B**–**F**). Orange circles represent AST I cohort. Violet circles represent AST II cohort. * *p* < 0.05, ** *p* < 0.01, *** *p* < 0.001.

**Figure 6 cells-14-00628-f006:**
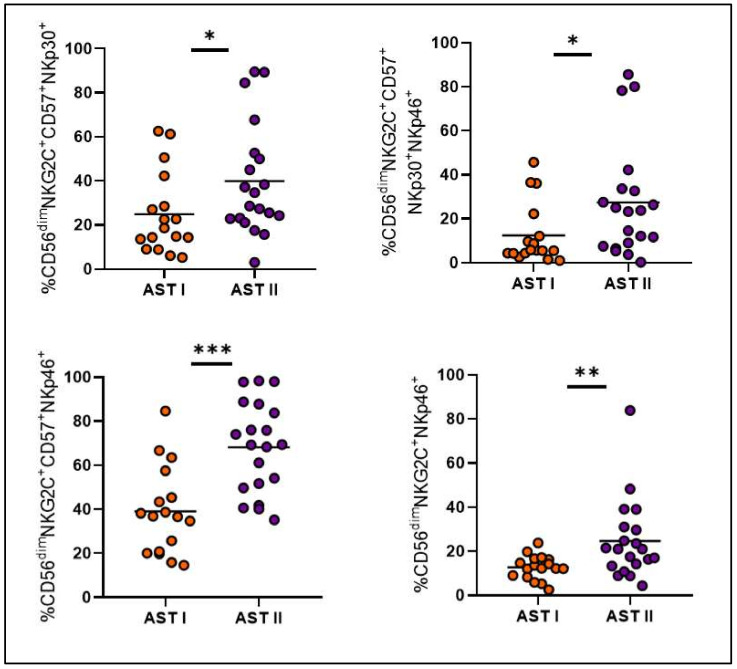
Comparison of adaptive-like NK cell subsets between both patient cohorts. Orange circles and bars represent AST I cohort. Violet circles and bars represent AST II cohort. * *p* < 0.05, ** *p* < 0.01, *** *p* < 0.001.

**Figure 7 cells-14-00628-f007:**
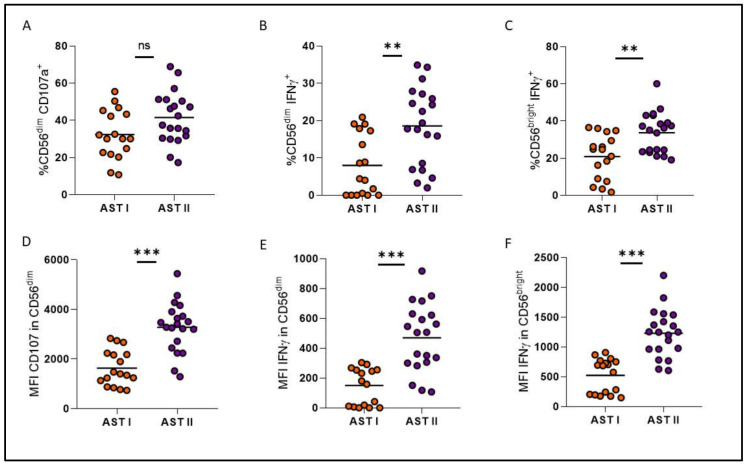
Comparison of degranulation (%CD107a) and IFNγ production in terms of proportion (**A**–**C**) and mean fluorescence intensity (MFI) (**D**–**F**) among CD56^dim^ and CD56^bright^ subsets between both patient cohorts. Orange circles represent AST I cohort. Violet circles represent AST II cohort. ns: non-significant, ** *p* < 0.01, *** *p* < 0.001.

**Figure 8 cells-14-00628-f008:**
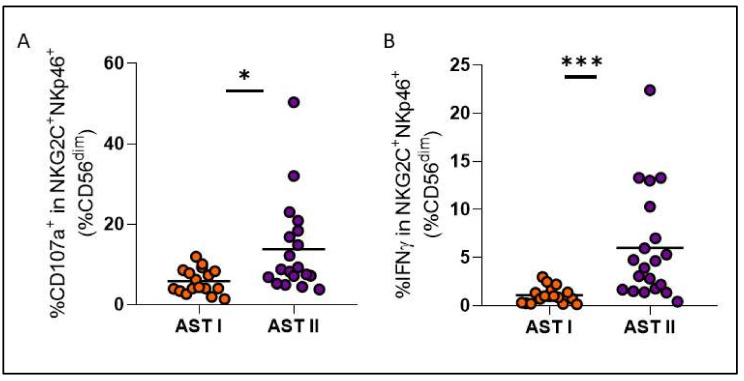
Comparison of degranulation (%CD107a) (**A**) and IFNγ production (**B**) among adaptive-like NK cells between both patient cohorts. Orange circles represent AST I cohort. Violet circles represent AST II cohort. * *p* < 0.05, *** *p* < 0.001.

**Figure 9 cells-14-00628-f009:**
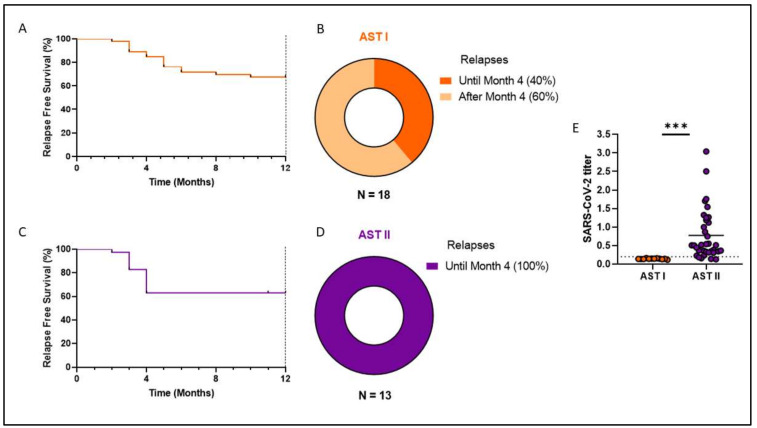
Relapse-free survival (**A**,**C**) and relapse rate (**B**,**D**) of both patient cohorts. Comparison of SARS-CoV-2 titer between both patient cohorts (**E**). (**E**) Orange circles represent AST I cohort. Violet circles represent AST II cohort. *** *p* < 0.001.

## Data Availability

The datasets generated and/or analyzed during the current study are available from the corresponding author upon reasonable request.

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
