# Peer review of "The Influence of the COVID-19 Pandemic in NK Cell Subpopulations from CML Patients Enrolled in the Argentina Stop Trial"

_cells, 2025, doi:10.3390/cells14090628_

Round 1
Reviewer 1 Report
Comments and Suggestions for Authors
Additionally, it would be interesting to examine differences in outcomes between patients who developed immunity through both infection and vaccination versus those who were vaccinated alone, if such data is available. Greater antigen exposure could potentially contribute to increased NK cell exhaustion, which may have implications for overall immune resilience, infection risk and immune surveillance.
Could the increased exhaustion of NK cells, as indicated by PD-1 expression in the AST II cohort, also be associated with a higher susceptibility to infections, particularly in the context of COVID-19-related immunosuppression if patients were exposed to it prior to the vaccination or COVID-19 related treatments. Given that steroids can exacerbate immune exhaustion, it would be valuable to investigate whether patients in the AST II cohort received COVID-related treatments and how these may have influenced immune function.
Author Response
Comment 1: Additionally, it would be interesting to examine differences in outcomes between patients who developed immunity through both infection and vaccination versus those who were vaccinated alone, if such data is available. Greater antigen exposure could potentially contribute to increased NK cell exhaustion, which may have implications for overall immune resilience, infection risk and immune surveillance.
Response 1: We appreciate this suggestion and agree that it would be valuable to explore differences in outcomes between patients who developed immunity through both infection and vaccination compared to those who were vaccinated alone. However, as of now, such data is not available in the current study. We recognize the importance of this analysis and recommend it as an avenue for future research once the necessary data becomes accessible. We added this statement in the discussion section (lines 233 – 238).
Comment 2: Could the increased exhaustion of NK cells, as indicated by PD-1 expression - n in the AST II cohort, also be associated with a higher susceptibility to infections, particularly in the context of COVID-19-related immunosuppression if patients were exposed to it prior to the vaccination or COVID-19 related treatments. Given that steroids can exacerbate immune exhaustion, it would be valuable to investigate whether patients in the AST II cohort received COVID-related treatments and how these may have influenced immune function.
Response 2: This is an interesting and valid point. While we agree that increased exhaustion of NK cells, as indicated by PD-1 expression, could potentially be linked to higher susceptibility to infections, particularly in the context of COVID-19-related immunosuppression, we currently do not have available data for all patients in the AST II cohort regarding prior exposure to COVID-19 or COVID-related treatments. Furthermore, we acknowledge that treatments such as steroids may exacerbate immune exhaustion, which could influence immune function. However, due to the lack of complete data on COVID-19 exposure and treatments for all patients in this cohort, we are unable to directly address this in the current study. This represents a significant area for future research.
Reviewer 2 Report
Comments and Suggestions for Authors
The work submitted by Sanchez and coworkers is an interesting study on the phenotypical differences observed in NK cells from CML patients recruited before and after COVID-19 pandemic. The study shows marked differences in NK cells between the two cohorts which seems to not be related to clinical differences but may be due to SARS-CoV-2 infection/vaccination, implying that pandemic could be taken into account when investigating for immunological biomarker. Although the study is quite brief and preliminary, it provides interesting hints for further research. However, some methodological imprecisions must be amended and a better discussion of obtained data is needed. Details below:
- Materials and methods should include the number of responsive/non responsive patients for each cohort and should also specify the criterion to define after how much time patients were defined as non-relapsed
- The authors performed several comparisons on NK cell-related markers between patient groups, but, based on what is reported in Materials and Methods, it seems that no correction has been applied and significant p-value has been kept at 0.05. This is not completely correct, since multiple tests increase the risk of spurious association. Therefore, the significant p-value should be lowered by Bonferroni’s correction at 0.003 (0.05/18 if only the showed variables were actually considered and there are not additional not shown variables). Bonferroni’s correction can be bypassed whether the obtained significant variables is higher than the number deriving from chance, but this must be demonstrated (for ref please see doi: 10.3389/fimmu.2022.811131)
- In my opinion, the statement in lines 176-180 should be a little bit revised. Indeed, the authors propose that NK cells in AST II cohort might have a not assessed, exhausted phenotype. By definition exhaustion means that a cell is no longer responsive, even when it receives proper stimuli, and NK cells from AST II patients appear normally active when challenged with K562 cells. The point is that K562 cells represent the gold standard for NK activation since the possess a very peculiar phenotype (lack of MHC-I molecules). Therefore, a more likely hypothesis is that, in AST II cohort, a change in NK cells occurred making them less able to recognize and kill residual tumor cells, but without reaching an exhaustion state, that would have been shown by a reduced capability to degranulate in response to K562 cells. In this context, I would suggest the authors to emphasise the phenotypical differences between K562 and CML immune phenotype, highlighting that K562-based assays might not completely recapitulate the response against other tumor cells
- For lines 203-207, I would recommend the authors to specify that HLA-E expression was up-regulated in lung cells
Author Response
The work submitted by Sanchez and coworkers is an interesting study on the phenotypical differences observed in NK cells from CML patients recruited before and after COVID-19 pandemic. The study shows marked differences in NK cells between the two cohorts which seems to not be related to clinical differences but may be due to SARS-CoV-2 infection/vaccination, implying that pandemic could be taken into account when investigating for immunological biomarker. Although the study is quite brief and preliminary, it provides interesting hints for further research. However, some methodological imprecisions must be amended and a better discussion of obtained data is needed. Details below:
Comment 1: Materials and methods should include the number of responsive/non responsive patients for each cohort and should also specify the criterion to define after how much time patients were defined as non-relapsed
Response 1: We appreciate this suggestion. Considering the importance of this information, we've added the following sentence in the Results section instead of Materials and Methods, which is highlighted in the revised manuscript (lines 102 – 108): “In AST I cohort, there were 18 relapsed patients, mostly within the first 6 months (13/18), with 2 relapses occurring between Month 6 and Month 12, 1 relapse at Month 21, and the remaining 2 relapses almost 3 and a half years after discontinuing treatment. Whereas for AST II cohort, there were 13 relapses, all within the first 4 months post-discontinuation. The original follow-up period for each cohort was two years, but it is worth noting that in the case of the first cohort, we have data on later relapses because most patients continued on molecular monitoring”.
Comment 2: The authors performed several comparisons on NK cell-related markers between patient groups, but, based on what is reported in Materials and Methods, it seems that no correction has been applied and significant p-value has been kept at 0.05. This is not completely correct, since multiple tests increase the risk of spurious association. Therefore, the significant p-value should be lowered by Bonferroni’s correction at 0.003 (0.05/18 if only the showed variables were actually considered and there are not additional not shown variables). Bonferroni’s correction can be bypassed whether the obtained significant variables is higher than the number deriving from chance, but this must be demonstrated (for ref please see doi: 10.3389/fimmu.2022.811131)
Response 2: We appreciate this suggestion. We have consulted with a statistician, and indeed, the most appropriate approach is to apply a correction to the obtained p-values. In this case, we have chosen the False Discovery Rate (FDR) correction using the Benjamini-Hochberg method, which controls the expected proportion of false discoveries among the significant results. This method offers a good balance between minimizing false positives and maintaining statistical power, making it particularly suitable for clinical studies with multiple measurements (in our case, several NK cell-related markers) (doi: 10.1191/0962280204sm363ra). We have clarified this in Materials and Methods section, lines 96 and 97. We have also added a supplementary table with all the originally obtained values ​​and their corresponding corrections (Supplementary Table 1). All figures that include comparisons between AST I and AST II have been reviewed and modified as appropriate, taking these corrections into account.
Comment 3: In my opinion, the statement in lines 176-180 should be a little bit revised. Indeed, the authors propose that NK cells in AST II cohort might have a not assessed, exhausted phenotype. By definition exhaustion means that a cell is no longer responsive, even when it receives proper stimuli, and NK cells from AST II patients appear normally active when challenged with K562 cells. The point is that K562 cells represent the gold standard for NK activation since the possess a very peculiar phenotype (lack of MHC-I molecules). Therefore, a more likely hypothesis is that, in AST II cohort, a change in NK cells occurred making them less able to recognize and kill residual tumor cells, but without reaching an exhaustion state, that would have been shown by a reduced capability to degranulate in response to K562 cells. In this context, I would suggest the authors to emphasise the phenotypical differences between K562 and CML immune phenotype, highlighting that K562-based assays might not completely recapitulate the response against other tumor cells
Response 3: We appreciate the reviewer’s suggestion. Indeed, while NK cells from the AST II cohort are able to degranulate and produce IFNy upon stimulation with K562 cells, we have observed that this response does not correlate with the clinical status of the patients. This discrepancy highlights the complexity of the immune response in the context of CML and suggests that other factors, such as the presence of specific ligands on leukemia stem cells (LSCs) or other tumor-associated factors, may play a role in regulating NK cell activity. However, as with our previous work (https://doi.org/10.3389/fimmu.2023.1241600), we acknowledge the limitations in measuring these ligands and their direct impact on NK cell function in the AST II cohort. Unfortunately, these specific ligands were not assessed in the current study, and further research would be needed to explore how they might influence NK cell responses in CML.
We clarified this point in the manuscript (lines 192 – 196), emphasizing that the degranulation and IFNy production of NK cells do not fully capture the complexity of their interaction with tumor cells and that additional factors should be considered in future studies.
Comment 4: For lines 203-207, I would recommend the authors to specify that HLA-E expression was up-regulated in lung cells
Response 4: We agree that it would be helpful to specify that HLA-E expression was up-regulated in lung cells, as this provides important context for the findings discussed in these lines. We added the proper clarification in line 224.
Reviewer 3 Report
Comments and Suggestions for Authors
This paper presents an immunological analysis of NK cell subpopulations in CML patients attempting treatment-free remission before and after the COVID-19 pandemic. The findings suggest significant immunological differences between pre- and post-pandemic cohorts, possibly linked to SARS-CoV-2 infection and/or vaccination.
The study explores a highly relevant topic, which is the potential impact of SARS-CoV-2 on NK cell function in a clinical setting. The findings have implications for understanding immune responses in hematological malignancies post-COVID.
The research effectively included 2 cohorts, laboratory test was selected properly, and the methods were described clearly. Statistical analysis was well executed, too.
Some of the weakness includes that:
- Small sample size (46 and 35 in each cohort), which may limit statistical power. The lack of significant differences in relapse-free survival between cohorts also suggests the need for larger validation studies.
- The study does not control for potential confounders, such as age, comorbidities, or COVID-19 vaccination types.
- While phenotypic changes in NK cells are well described, direct functional assays (e.g., killing assays with primary leukemia cells) would strengthen the findings.
- In Figure 2 and Figure 5, the data are too complex. Clearer legends and/or annotation may help.
- The discussion can be more structured by clearly separating finding in different sections related to NK cell phenotype, function, and clinical outcomes.
Author Response
This paper presents an immunological analysis of NK cell subpopulations in CML patients attempting treatment-free remission before and after the COVID-19 pandemic. The findings suggest significant immunological differences between pre- and post-pandemic cohorts, possibly linked to SARS-CoV-2 infection and/or vaccination.
The study explores a highly relevant topic, which is the potential impact of SARS-CoV-2 on NK cell function in a clinical setting. The findings have implications for understanding immune responses in hematological malignancies post-COVID.
The research effectively included 2 cohorts, laboratory test was selected properly, and the methods were described clearly. Statistical analysis was well executed, too.
Some of the weakness includes that:
Comment 1: Small sample size (46 and 35 in each cohort), which may limit statistical power. The lack of significant differences in relapse-free survival between cohorts also suggests the need for larger validation studies.
Response 1: Thank you for your valuable comment. We agree that the sample size (46 and 35 in each cohort) could limit the statistical power of the study. Unfortunately, these sample sizes reflect the available patient population. However, we believe that the findings still provide valuable insights into the potential impact of SARS-CoV-2 on NK cell function in CML patients, which could serve as a basis for future larger-scale studies.
We also agree with the reviewer that the lack of significant differences in relapse-free survival between cohorts highlights the need for validation in larger cohorts. We addressed this limitation in the manuscript and recommend further research to validate our findings (lines 238 – 244).
Comment 2: The study does not control for potential confounders, such as age, comorbidities, or COVID-19 vaccination types.
Response 2: Thank you for your thoughtful comment. We agree that controlling for potential confounders such as age, comorbidities, and COVID-19 vaccination types is important. However, we have already addressed age as a potential confounder by comparing it between cohorts, and our analysis showed no significant age-related differences that could explain the observed immunological changes.
Regarding comorbidities, we acknowledge that they may represent an important confounder; however, due to the complexity and variability in the types and severity of comorbid conditions across patients, and given the small sample size, it was not feasible to include them as a controlled variable in this study.
As for the COVID-19 vaccination types, we recognize the potential influence of different vaccine types on immune responses. However, due to the lack of complete data on vaccination types for all patients, we were unable to assess their effect. We addressed this limitation in the manuscript (lines 233 – 238).
Comment 3: While phenotypic changes in NK cells are well described, direct functional assays (e.g., killing assays with primary leukemia cells) would strengthen the findings.
Response 3: Thank you for this important suggestion. We agree that direct functional assays, such as killing assays with primary leukemia cells, would provide valuable additional insights and strengthen our findings. However, we would like to emphasize that patients in discontinuation, having been under treatment for a long time and expressing very low or even undetectable levels of the chimeric oncogene BCR-ABL, do not have a sufficient number of residual leukemic stem cells (LSCs) in circulation in order to be able to detect, characterize and isolate them to perform this type of assays. We addressed this issue in the manuscript (lines 194 - 196). Nonetheless, we believe that the phenotypic changes observed in the NK cells still offer meaningful information.
Comment 4: In Figure 2 and Figure 5, the data are too complex. Clearer legends and/or annotation may help.
Response 4: Thank you for this suggestion. We have reorganized all the figures so that the data can be presented more clearly.
Comment 5: The discussion can be more structured by clearly separating finding in different sections related to NK cell phenotype, function, and clinical outcomes.
Response 5: Thank you for this comment. We have rewritten the discussion to make it easier to follow (lines 185 – 201) and have also added limitations that we believe are valuable for the correct interpretation of the conclusions we can obtain from this study (lines 233 – 244).
Round 2
Reviewer 2 Report
Comments and Suggestions for Authors
The authors addressed all my points. The manuscript can be accepted in the current form